# Optimization of running-in surface morphology parameters based on the AutoML model

**Guangyuan Ge[1], Fenfen Liu[2], Gengpei Zhang** [1]*

**1** School of Electronics and Information, Yangtze University, Jingzhou, Hubei, China, **2** School of Computer Science, Yangtze University, Jingzhou, Hubei, China

* judgebill@126.com

## Abstract

Running-in is an important and relatively complicated process. The surface morphology prior to running-in affects the surface morphology following the running-in process, which in turn influences the friction and wear characteristics of the workpiece. Therefore, the establishment of a model for running-in surface morphology prediction is important to investigate the process and optimize the surface design. Black-box models based on machine learning have robust complex object simulation performance. In this paper, five common machine learning methods are applied to establish running-in modeling performance based on surface morphology parameters. The support vector machine has the best model performance. The change law of the surface morphology parameters is obtained based on model testing, and the surface morphology optimization is explored. When better oil storage capacity is required, the recommendation is to increase the Sq, Sdq and Sk surface parameter values while setting medium Sdc and Sdr surface parameter values. When a lower coefficient of friction (COF) is required, Sdc and Sdr should be decreased, and Sq and Sdq should be increased. When better support performance is required, Sdc, Sdq, and Sdr should be increased. This article provides a solution to establish a link between surface design and functional performance in the steady wear stage, further filling the gap in quality monitoring of lifecycles.

## 1. Introduction

Running-in is an important wear stage that a friction pair undergoes in the early operational stage. This stage has an important impact on workpiece wear characteristics, and the process is relatively complex [1]. The improvement of running-in control is helpful to extend the lifespan of the mechanical system and ensure stable operation of the system. Establishing a suitable wear model is the key to achieving this goal.

Given the input, intermediate state characterization, and the result of the running-in process, the surface morphology changes significantly [2,3]. Surface morphology after running-in shows many correlations with many functional performances, and unworn surface

**Funding:** The author(s) received no specific funding for this work.

**Competing interests:** The authors have declared that no competing interests exist.

morphology is related to workpiece processing means. Establishing the relationship of surface morphology before and after running-in is significant for optimizing the process.

Many running-in studies have researched the influence of surface morphology on the related characteristics of friction pairs. As a directly involved factor, the surface morphology of friction pairs [4,5] shows more information than that of wear debris [6]. Usually, studies are carried out based on surface morphology parameters instead of surface morphology [7], and surface profile roughness is used to predict the surface roughness [8] and wear rate [9] and evaluate the degree of wear [10]. Moreover, the surface roughness based on the surface 2D profile shrinks the surface morphology information.

Current running-in wear research is either focused on obtaining approximate laws through experiments or on explaining the phenomenon of surface modification by simplifying the running-in phenomenon and analytical models [11–13]. However, these methods cannot describe the laws of surface morphology changes before and after running-in. Considering the numerous interactive factors that are involved, the complexity of the running-in process cannot be explained simply [1,14]. However, black-box methods can provide more factor detail in running-in wear than conventional methods. Considering its effectiveness and wide use, the machine learning (ML) method is an inviting vantage point from which to explore the surface morphology change laws and obtain the unknown optimal running-in surface parameters.

ML is mainly employed to analyze sample data, discover patterns, and establish models. These models are then utilized to predict unknown or unobservable sample data [15]. Many ML methods are commonly used, including artificial neural networks (ANNs) [16,17], support vector machines (SVMs) [18,19], tree-based pipeline optimization tools (TPOTs) [20], bootstrap aggregation (bagging) [21,22], and random forests (RFs) [23].

In 2015, to simulate the surface profile roughness of materials, Agrawal et al. [24] established multiple regression, random forests, and quantile regression models. In 2017, Wu, D et al. used RFs and support vector regression for tool wear prediction [25] and used RFs to predict tool wear in dry milling operations [26]. To distinguish the difference of ML techniques for predicting tool wear during milling, Wu et al. [27] compared the performance of ANN, SVM regression, and RF. In 2018, Bustillo et al. [28] used artificial regression trees, multilayer perceptrons, radial basis networks, and RFs to predict the surface roughness, quality loss, and flatness deviation of machined surfaces under different processing parameters [29]. To achieve real-time prediction of surface roughness deviation, Pimenov et al. [30] evaluated ML methods such as the RF, regression trees, the standard multilayer perceptron, and the radial basis function. Gouarir A et al. proposed a tool wear prediction system based on the convolutional neural network method [31]. In 2019, to realize real-time and accurate monitoring of tool wear during processing, Kong D et al. used traditional methods such as an ANN and an SVM to achieve tool wear prediction and then proposed tool wear prediction models based on core major component analysis and related vector machines [32]. Altay O et al. used linear regression, an SVM, and Gauss process regression to predict wear quantities [33]. In 2020, Thankachan T et al. predicted and analyzed the dry sliding wear rate on novel copper-based surface composites based on the source feed forward back propagation model of ANNs [34]. Cheng M et al. established an intelligent prediction model by using the support vector regression method [35]. In 2021, Alajmi et al. used Gaussian process regression, an SVM and an ANN to establish a cutting force prediction model of alloy steel [36].

Many methods based on ML have been widely applied in the field of tool wear prediction. Some researchers have used various ML methods to predict surface roughness, wear rate and wear volume in wear research. However, ML methods are barely used to study the changing laws of the surface morphology in the running-in wear process. Previous studies have shown

that ML methods achieve great superiority in the field of wear. Thus, if ML methods can be used to conduct research in the field of running-in wear, it will greatly reduce the complexity of the research and improve the prediction accuracy.

Based on the analysis above, the main ML methods are applied in this paper. First, the training data of the model come from the running-in wear experiment, and the detailed description is provided in sections 3.1 and 3.2. Then, surface parameter correlation analysis is conducted for the dimension reduction of model input/output, and the selection results are shown in section 4.2. According to the theory described in sections 2.1–2.5, numerous models are established in section 4.1 and evaluated in section 4.3 for method selection. To understand the influence of surface morphology in running-in, the optimal model based on the selected method is tested, the detailed operation is presented in section 4.4, and the results are shown in sections 5.1 and 5.2. To verify the model and testing result, extra experimental data based on new working conditions are obtained. The data acquisition and verification are shown in section 5.3. Finally, through the influence between the parameters, the optimization guidance of surface morphology parameters in practice is presented in section 5.4.

Based on the sequence mentioned above, this paper attempts to establish linkages between the design of unworn surfaces and the performance of worn surfaces after running-in.

## 2. Methods

### 2.1. ANN

The ANN is a brain-type intelligent system that simulates the brain basic unit structure and function. It is a nonlinear dynamic system composed of many interconnected neurons. These neurons are connected, thus forming an ANN. A neuron receives the output from other neurons as its input. Then, it distributes its output to other neurons. Interconnections between these neurons are weighted and used to regulate neuron interactions [37]. The ANN algorithm has high accuracy and is suitable for noisy data sets. However, this method has weak interpretation and many parameters that are relatively difficult to adjust.

ANNs have strong adaptation capabilities in many model establishments of complex problems and are used in various fields [16]. Many researchers use ANNs for the prediction of wear, wear resistance, wear depth, wear rate, and flank wear [38–41]. Hanief [8] established an ANN model to research the modification of the surface morphology roughness over time during the running-in wear period. Argatov [42] established a neural network support regression model to predict the wear rate.

### 2.2. Bagging

As a group learning algorithm of ML, bagging was first introduced by Leo Breiman [21]. The main idea is to obtain different subtraining sets by sampling from the original sample data set. Then, subtraining sets are employed to obtain different classifiers and combine them in various ways to obtain the final classifier. The idea of using repeated random sampling techniques to select the training set in the bagging algorithm enhances the difference between learning machines. As such, the generalization ability and prediction accuracy of the learning algorithm is improved [43]. In the financial field, bagging is often employed to address time series data, which occasionally presents low prediction accuracy [44,45].

### 2.3. RF

The accuracy of RF wear prediction has already been proven [24,46]. To train and predict samples, RF constructs a classifier based on multiple trees, first seen in the work of Leo Breiman

[23]. Each classifier determines the optimal classification result through voting, and the output category is determined by the individual tree. Due to the internal decision trees, RF has insensitivity to outliers and high accuracy. The generalization error of the algorithm converges with the growth of the tree number, thus increasing the difficulty of passing the fitting. In addition, fewer candidate features need to be considered when dividing each tree, the calculation speed is relatively fast, and the importance of variables can be estimated. The RF method has high prediction accuracy in research fields such as tool wear prediction [47], conveyor wear rate prediction [48], gear failure detection [49], and erosion prediction [50].

## 2.4. SVM

An SVM is a modern contribution to data-driven model establishment [51,52]. The SVM is a binary classification model, which makes it suitable for small sample learning. Based on diverse kernel functions, an SVM minimizes structural risk to model high-dimensional problems. It works by finding a dividing hyperplane. The goal is to maximize the dividing interval and turn it into a traditional routine—a convex quadratic programming problem. Based on SVMs, researchers have established a surface morphology prediction model after the running-in process [53] and a tool wear prediction model [54,55], which have been successfully applied in various fields [18].

## 2.5. TPOT

TPOT is an automatic machine learning (AutoML) framework developed by the Genetic Algorithm Laboratory of the University of Pennsylvania [20]. It can find the most suitable algorithm and its parameters for the current data situation from thousands of possibilities [56]. Furthermore, it is conducive to achieving ML pipeline design automation. Compared with basic ML analysis, TPOT performance is superior [57,58].

TPOT can be employed to explore a learning pipeline composed of any combination of selectors, converters, and estimators. To enhance the scalability and interpretability, a template for the searched pipeline can be specified, and a selector of the feature set can be appended as the input preprocessing port for every pipeline to separate the model training data by different small feature sets, thus allowing genetic programming to seek the best performance pipeline [59]. TPOT has been successfully applied in biomedicine [59,60].

In summary, comparison of five AutoML methods mentioned above is shown in Table 1.

**Table 1. Comparison of automl methods.**

|  | Advantage | Disadvantage | Feature |
|---|---|---|---|
| ANN | Parallel distribution, strong processing power | Excessive parameters | A parameter set with a massive amount of applicable data and an internal connection between the parameters |
|  | Approximate complex nonlinear relationships | Difficult to interpret |  |
| SVM | High-dimensional features problems for small samples | Seeking suitable kernel function | Suitable for parameter sets with a large feature space and can handle nonlinear features |
|  |  | Sensitive to missing data |  |
| TPOT | Automation pipeline structure design | Not friendly to natural language processing | Time consuming |
| RF | Variables' priority, anti-interference ability | Overfits in some noisy classification and regression scenarios | Typical applications in classification of various images and data, face detection |
| Bagging | Helps reduce differences, avoid overfitting | Less interpretability | Suitable for multiclassification and regression |

## 3. Experimental investigation

### 3.1. Morphology acquisition

Surface morphology before and after running-in was obtained via a surface comprehensive measuring instrument Form Talysurf PGI830, which is shown in Fig 1. This instrument uses the contact probe to scan the surface morphology and produces laser interference through probe displacement. Hence, surface morphology is obtained.

The transverse measurement range of PGI830 is 200 mm, the longitudinal measurement range is 0–8 mm, and the measurement accuracy is 0.8 nm. During the running-in experiment of training samples, the measurement area of all surface morphologies was 1 mm$^2$, and the measuring interval was 10 μm. After each surface measurement, plane fitting was used to remove the shape error. Then, a robust Gaussian filter was used to separate the waviness and roughness, and the cutoff wavelength of the filter was 0.25 mm. Finally, the surface morphology of a pin head was described based on the surface evaluation parameters of the region method.

In this experiment, two types of surface morphologies with different textures were used: one with isotropic multimodal structure and the other with multidirectional groove texture. The surface morphology height of each texture can be divided into four types. The initial morphology of four types of surfaces is shown in Fig 2.

### 3.2. Experimental design

The running-in wear process was established on a universal friction and wear testing machine, which was completed by the pin disc friction pair. The universal friction and wear tester wwm-1 is shown in Fig 3. Different working conditions were simulated by lever loading, and wear was achieved by the motor driving the rotary pin on the fixed plate. The working range of the test force is 0–1000 N, the measuring range of friction torque is 0–2500 N·mm, the temperature control range is 20˚C - 300˚C, the spindle speed range is 10–2000 r/min, and the time control range is 1 s-9999 min. The lower friction plate is loaded through the lever, and the

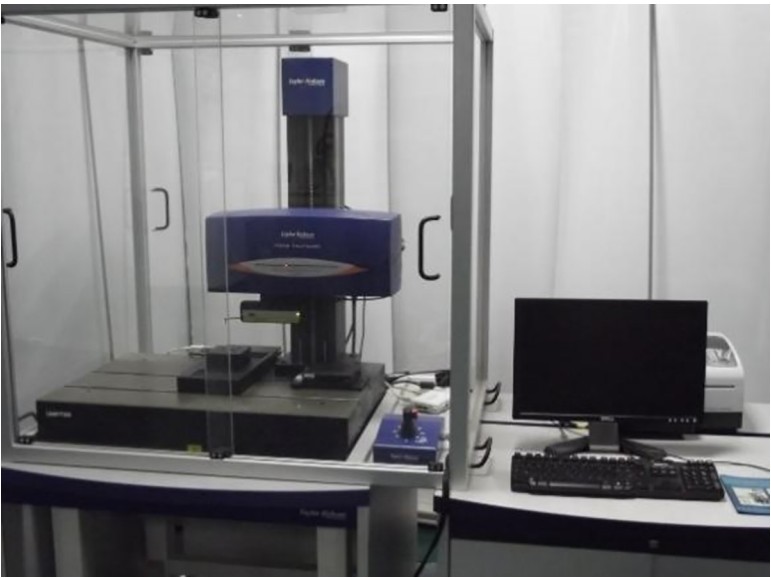

**Fig 1. Surface morphology instrument PGI830.**

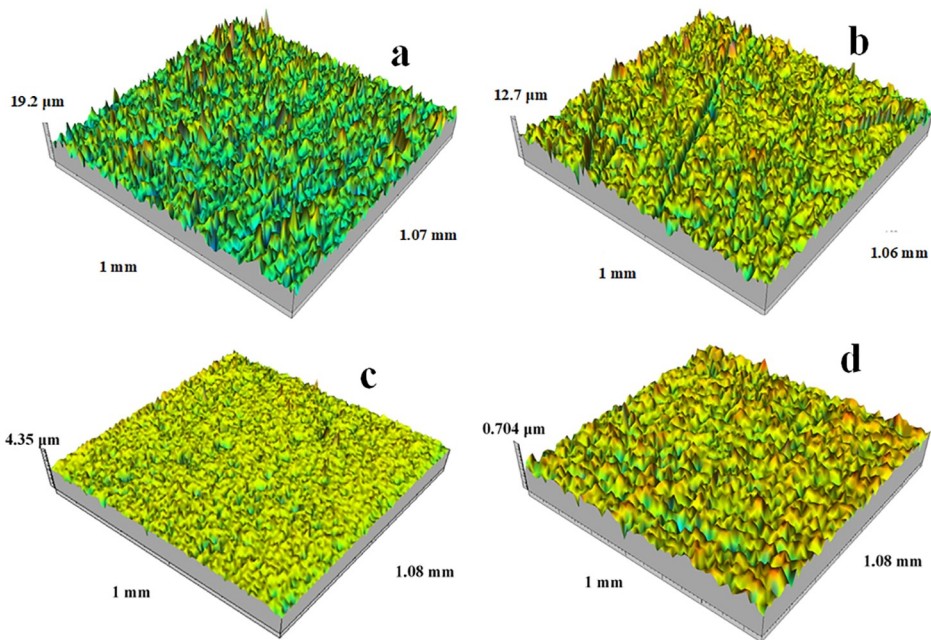

**Fig 2. Initial morphology of four types of surfaces.** (a) the multimodal surface with larger height. (b) Multidirectional grooved surface with a larger height. (c) the multimodal surface with smaller height. (d) Multidirectional grooved surface with a smaller height.

upper motor rotates the pin head fixture to achieve circular sliding of the upper pin head on the friction plate.

In the experiment, the pinhead material was 45 # steel (ASTM1045), which was processed into a cylindrical shape. The diameter of the pin was 0.3 cm, while the length of the pin was 1.7 cm. Following the heat treatment, its hardness reached 31–36 HRC. The material of the

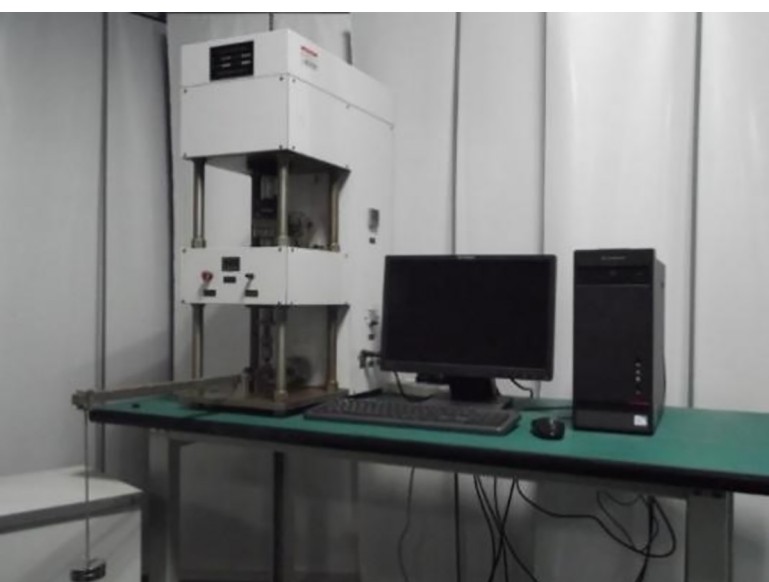

**Fig 3. Universal friction and wear tester wwm-1.**

friction disc was bearing steel (ASTME52100), which was processed into a disc shape. The diameter of the disc was 5.4 cm, while the thickness of the disc was 1 cm. Following the heat treatment, its hardness reached 59–62 HRC. The sliding friction of the pin disc friction pair was infiltrated in the lubricant. A variety of lubricants were used. For example, one of the employed lubricants was diesel engine lubricating oil, whose density is 0.886 g/cm$^3$ and kinematic viscosity is 80 mm$^3$/s at 20˚C.

Within the experiment, to simulate the common running-in wear process, the sliding speed was set low, and the load was set small. The load (①20 N, ②30 N, ③40 N), sliding speed (①20 r/min, ②40 r/min, ③60 r/min) and lubricant viscosity (①10 mm3/s, ②80 mm3/s, ③130 mm3/s) were employed according to three different levels of orthogonal design, thus obtaining nine different running-in conditions. In this experiment, four types of surface morphologies and nine different running-in conditions were combined, thus resulting in a total of 36 sets of different generated surface morphology sample parameters. Different sliding speeds and sliding times were employed to ensure that all samples eventually reached the same sliding distance (240 circles). The COF of all wear experiments reached stability before the sliding distance finished.

## 4. Model establishment and testing

### 4.1. Data manipulation

Among the five ML methods mentioned in the introductory section and for a specified surface morphology parameter after running-in, TPOT can select the most suitable ML algorithm and optimize its parameters through a genetic algorithm. The remaining four ML methods can build the best model through parameter optimization in the given algorithm.

The modeling process of running-in wear based on ML is as follows:

1. Training data of running-in wear is imported.

2. The experimental data are preprocessed by different ML methods.

3. Divide the training set and the validation set at an 11:1 ratio.

4. Modeling the surface parameters is accomplished after running-in by five ML methods.

5. Evaluate the quality of the model.

6. The corresponding prediction data can be obtained by inverse processing of the model output data.

7. Further evaluation of model quality is performed based on prediction data and experimental data.

8. Check the finish of 200 experimental process. If it is finished, go to step 9. If it is not finished, go to step 4.

9. Calculate the average value of the prediction output of the 200 experimental processes.

10. Check the finish of validation of all divided data set combinations. If it is finished, go to step 11. If it is not finished, go to step 3.

12. The best model can then be employed for subsequent testing.

According to the process shown in Fig 4, a surface morphology parameter prediction model after running-in can be established according to the following training method:

$$(S_{1T}^*, S_{2T}^*, \ldots\ldots, S_{nT}^*) = \mathrm{Model}_S((W_{1T}, W_{2T}, \ldots\ldots, W_{nT}), (S_{1T}', S_{2T}', \ldots\ldots S_{nT}')) \qquad (1)$$

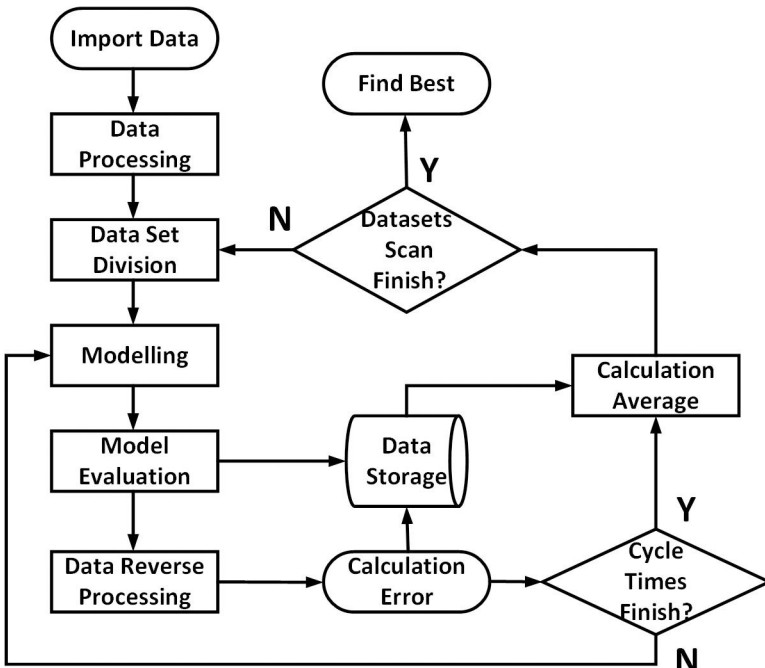

**Fig 4. Modeling process of the running-in wear prediction model based on ML.**

where $(S_{1T}^*, S_{2T}^*, \ldots\ldots, S_{nT}^*)$ is the surface parameter of the experimental output data for training the surface parameters, $(S_{1T}', S_{2T}', \ldots\ldots S_{nT}')$ is the surface parameter of the experimental input data for training, and $(W_{1T}, W_{2T}, \ldots\ldots, W_{nT})$ is the working condition parameter of the experimental input data for training.

Based on the model $Model_S$ established by employing different ML methods, surface parameters after running-in can be predicted based on the required working conditions and surface parameters before running-in.

$$(S_{1P}^*, S_{2P}^*, \ldots\ldots, S_{nP}^*) = Model_S((W_{1P}, W_{2P}, \ldots\ldots, W_{2P}), (S_{1P}', S_{2P}', \ldots\ldots S_{nP}')) \qquad (2)$$

where $(S_{1P}^*, S_{2P}^*, \ldots\ldots, S_{nP}^*)$ is the result of surface morphology parameters predicted by ML after data preprocessing. This result can become the prediction results of surface morphology parameters $(S_{1P}, S_{2P}, \ldots\ldots, S_{nP})$ by employing inverse data processing. $(W_{1P}, W_{2P}, \ldots\ldots W_{nP})$ is the working condition parameter of surface morphology after data processing, and $(S_{1P}', S_{2P}', \ldots\ldots, S_{nP}')$ is surface morphology parameters before running-in and after data preprocessing.

According to general training methods of the ML model, training and prediction methods of running-in wear surface morphology based on five types of AutoML are as follows.

The establishment of prediction model based on AutoML is shown in Fig 5. First, based on the surface morphology and working conditions of the friction pair, running-in wear experiments are carried out. Based on experimental data, surface parameters before and after running-in can be obtained. When modeling by AutoML, these parameters are used as inputs and outputs. Furthermore, the running-in wear prediction model is obtained by training five ML algorithms that are controlled by the AutoML method. Finally, based on the trained model, worn surface parameters after running-in can be predicted from known initial surface parameters without a running-in process. By analyzing the difference between the predicted output and the experimental output, the optimal model can be found.

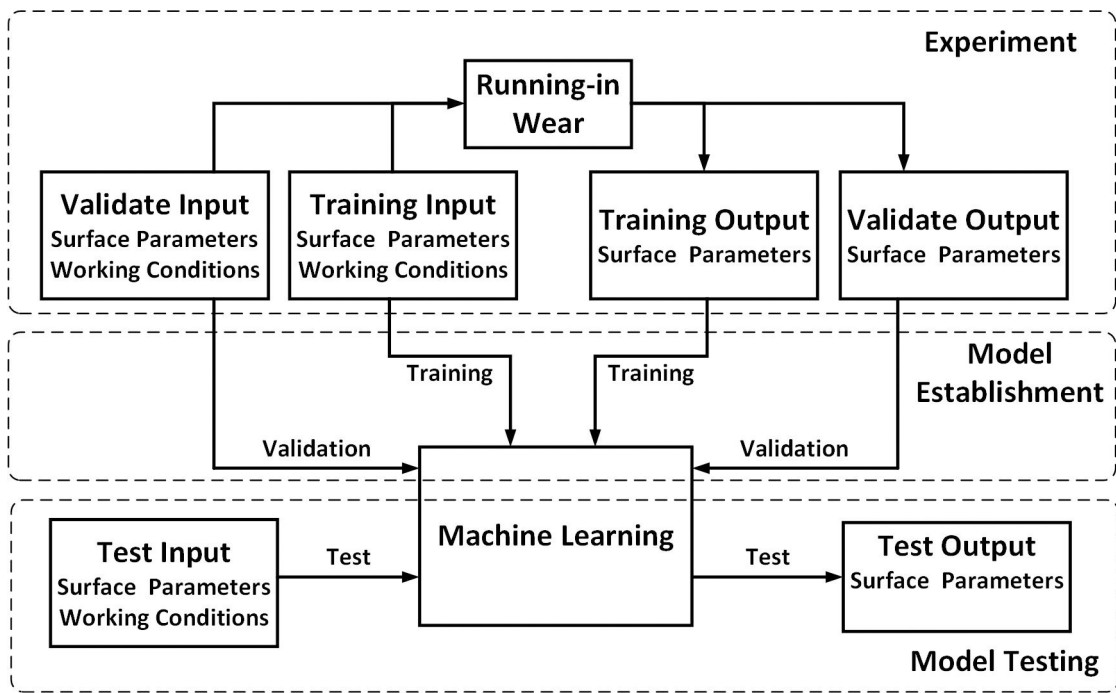

**Fig 5. Establishment of running-in wear prediction model based on AutoML.**

To test the established model, surface parameters to be studied after running-in have to be first determined. Then, to obtain the change law of surface parameters after running-in, surface parameters before running-in are deliberately constructed.

## 4.2. Parameter selection

A total of 36 sets of data samples were collected in this experiment, with a total of 33 surface morphology parameters according to areal surface characterization. The height parameters include Sa, Sq, Sku, Ssk, Sp, Sv, and Sz. Spatial parameters include Sal, Str, and Std. Hybrid parameters include Sdq, Sdr, Ssc, and Sds. The functional parameters include Sdc, Spk, Svk, Sk, Sbi, Sci, Svi, Sr1, and Sr2 [61].

The correlation analysis of 33 surface morphology parameters obtained by experiments was carried out by Pearson correlation coefficient analysis. Finally, five parameters with the strongest correlation are found for modeling and testing. Correlation chart of five parameters is shown in Fig 6.

The root mean square deviation Sq includes the global height parameter of the measured surface morphology, and the surface core roughness height Sk mainly represents the support performance of the running-in surface. This includes the support height of the measured surface morphology. Both Sq and Sk provide key height information on surface morphology during running-in. The root mean square slope Sdq can be indirectly utilized to obtain the surface support performance and wear surface height. By characterizing Sdq and Sdr, surface texture and surface height information can also be indirectly obtained [62].

Based on the above description, Sq, Sdc, Sdq, Sdr, Sk, velocity, viscosity, and load are taken as input parameters. Parameters $Sq^*$, $Sdc^*$, $Sdq^*$, $Sdr^*$, and $Sk^*$ are taken as output parameters for modeling and testing.

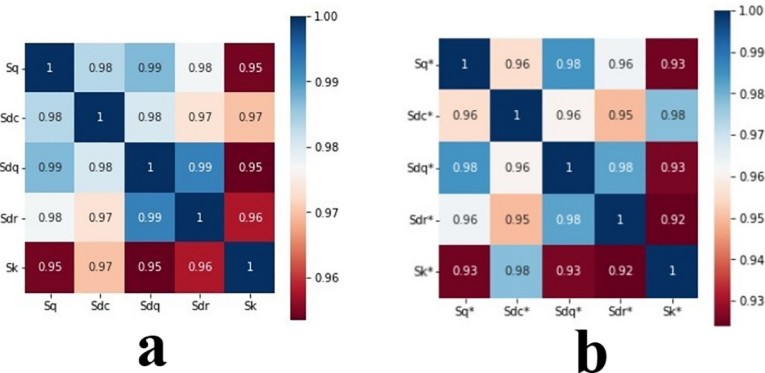

**Fig 6. Parameter correlation chart.** (a) Correlation before running-in. (b) Correlation after running-in.

## 4.3. Method selection

The model is evaluated with the average value of the absolute error (MAE) and coefficient of determination ($R^2$) of the data. MAE has the general expression:

$$MAE = \frac{1}{n}\sum_{i=1}^{n}|\widehat{y}_i - y_i| \tag{3}$$

When MAE is equal to 0, it means that the model's predicted value is entirely consistent with the experimental value. With an increase in MAE, the error of the model is also increased. This, in turn, decreases the effectiveness of the model.

The parameter $R^2$ can be expressed as:

$$R^2 = 1 - \frac{\sum_i\widehat{(y_i - y_i)}^2}{\sum_i(\bar{y}_i - y_i)^2} \tag{4}$$

As $R^2$ approaches 1, the accuracy of the model is increased as well.

Table 2 shows the evaluation of the modeling error and model quality of five surface parameters under different AutoML conditions. For different parameters, applicable machine learning methods are also different. For parameter Sq, the TPOT modeling results are the best (error of 0.74%), followed by SVM (error of 1.13%). For parameter Sdc, TPOT modeling results are the best with an error of 1.33%, followed by SVM with an error of 1.41%. For parameter Sdq, SVM is the best with an error of 1.22%, followed by RF and the corresponding error of 1.63%. For parameter Sdr, TPOT is the best (error 1.24%), followed by SVM (error 1.26%). For parameter Sk, TPOT is the best (error 0.74%), followed by ANN with an error of 2.07%. The combined MAE and $R^2$ results show that TPOT has the best modeling effect, followed by the SVM.

**Table 2. Modeling effect of different automl algorithms.**

|  | ANN | | SVM | | TPOT | | Bagging | | RF | |
|---|---|---|---|---|---|---|---|---|---|---|
|  | MAE | $R^2$ | MAE | $R^2$ | MAE | $R^2$ | MAE | $R^2$ | MAE | $R^2$ |
| Sq | 1.69% | 0.9868 | 1.13% | 0.9972 | 0.74% | 0.9861 | 4.60% | 0.9669 | 1.32% | 0.9903 |
| Sdc | 2.70% | 0.9870 | 1.41% | 0.9816 | 1.33% | 0.9972 | 7.65% | 0.9618 | 1.47% | 0.9858 |
| Sdq | 3.27% | 0.8173 | 1.22% | 0.9846 | 1.77% | 0.9979 | 7.67% | 0.8457 | 1.63% | 0.9873 |
| Sdr | 2.17% | 0.9949 | 1.26% | 0.9994 | 1.24% | 0.9885 | 8.37% | 0.9625 | 4.03% | 0.9893 |
| Sk | 2.07% | 0.9894 | 3.69% | 0.9369 | 0.76% | 0.9990 | 8.93% | 0.7129 | 4.42% | 0.9384 |

The POT results show that the SVM is used to model Sq, Sdq and Sdr, elastic net is used to model Sdc, and the stochastic gradient descent regressor is used to model Sk. The combined error size and model evaluation findings shown in Table 2 indicate that the SVM is the best method to model the five surface parameters required in this paper. This is consistent with the research results of other authors [63]. Therefore, to ensure the consistency of the running-in process research, the SVM is employed for test modeling in this paper.

For investigations regarding Sdc and Sk as running-in parameters, conclusions obtained by TPOT in this paper can be used for reference. The SVM is recommended as the modeling method when comprehensive parameter modeling analysis is required.

At the same time, ML approaches are learning methods that rely on data. ML mainly analyzes sample data, discovers laws from data and builds a suitable model. This model can be used to predict unknown sample data and unobservable sample data. Theoretically, the unique advantages of the SVM show better modeling effects in the process of running-in wear modeling: 1. the process of an SVM searching for support vectors is a convex optimization problem; 2. An SVM transforms a nonlinear problem into a linear problem through a kernel function, which can weaken the complexity of the original problem; 3. The SVM extracts support vector data from the training data and performs weight processing on the support vector data. If the amount of training data is small, all training data can be used as support vectors to build the model, which greatly reduces the number of training samples. Furthermore, SVMs use the principle of structural risk minimization, which makes the model stronger in generalization.

## 4.4. Model testing

Based on the analysis of section 4.3, after finding the suitable modeling method, surface parameters are modified to construct testing data by simulating the experimental data. Then, the testing data are input into the established model. According to exported results, the running-in parameters of the testing data space are explored to analyze the influence of surface morphology in running-in.

Data distribution analysis is carried out on five groups of input surface parameters. According to Fig 7, the distribution of experimental data is uneven. If all data are input into the model

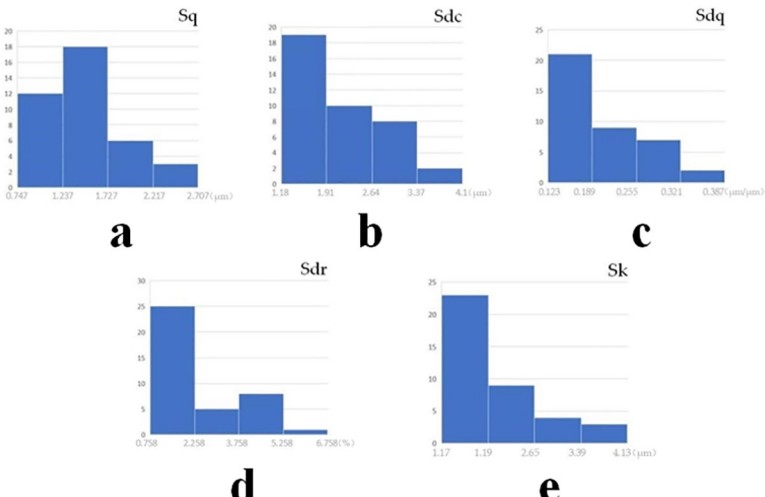

**Fig 7. Analysis of data distribution.** (a) Data distribution of Sq of unworn surface. (b) data distribution of Sdc of unworn surface. (c) Data distribution of Sdq of unworn surface. (d) Data distribution of Sdr of unworn surface. (e) Data distribution of Sk of unworn surface.

**Table 3. Data range of surface parameters.**

| Surface parameters | Experiment data range | Testing data range |
|---|---|---|
| Sq | 0.747 μm– 2.45 μm | 0.747 μm– 1.68 μm |
| Sdc | 1.18 μm– 3.6 μm | 1.18 μm– 2.76 μm |
| Sdq | 0.123 μm/μm– 0.343 μm/μm | 0.123 μm/μm– 0.236 μm/μm |
| Sdr | 0.758% - 5.73% | 0.758% - 2.75% |
| Sk | 1.17 μm– 3.85 μm | 1.17 μm– 2.59 μm |

for prediction, a relatively large deviation can occur between the predicted and experimental values in the range with many fewer samples. Therefore, when more prediction accuracy is desired, the testing data range should be reduced. The data range of surface parameters is shown in Table 3.

Within the decreased testing data range, the data are divided into 10 equivalent range parts and input into the optimal prediction model. To construct suitable testing input data, working condition parameters are set as experimental data, one of the surface parameters is selected to test the range division, and the other four surface parameters are set at an experimental value approximating range median.

To investigate the influence law of a single parameter, the effect of a single parameter changes before running-in on Sq after running-in is taken as an example. The Sq parameter before running-in increases according to the equivalent value, while other parameters before running-in remain unchanged and are set to the experimental value close to the median value of this parameter. The variation in Sq after running-in with an increase in Sq before running-in is recorded. From this, to analyze the independent influence of selected morphology parameters of the initial surface on the selected single morphology parameter of the worn surface, 25 tests are carried out.

Surface morphology parameters after running-in are affected by the two most relevant surface morphology parameters before running-in, which is similar to the research on the influence of a single parameter. For this case, two parameters are set to increase by the same value. Finally, a data mesh is obtained to analyze a running-in parameter.

## 5. Results and discussion

In the following discussion, Sq, Sdc, Sdq, Sdr, and Sk are surface parameters before running-in, and $Sq^*$, $Sdc^*$, $Sdq^*$, $Sdr^*$, and $Sk^*$ represent surface parameters after running-in.

### 5.1. The result of a single characteristic parameter

As shown in Fig 8, the curve line chart shows the change rate of the parameters after running-in. For example, the orange line in represents the growth rate of $Sq^*$ when only Sdc increases by an equivalent amount within the modified data range (Fig 8A). The scatter diagram in Fig 9 shows the change degree of the parameters after running-in that is affected by the parameters before running-in within the modified data range.

The growth rates of the five parameters after running-in gradually increase with increasing Sq. Among them, the values of $Sdc^*$ and $Sk^*$ decrease at the beginning. When Sdc is increased by the same amount, the change rates of the five parameters after running-in increase and then decrease. In the process of Sdq growth, the change rates of $Sq^*$, $Sdr^*$, and $Sdq^*$ increase and then decrease slightly, while the change rates of $Sdc^*$ and $Sk^*$ always increase. When Sdr's value increases, the change rates of $Sq^*$, $Sdr^*$, and $Sdq^*$ first increase and then slightly decrease, while the change rates of $Sdc^*$ and $Sk^*$ always increase. The impact trends of Sk and Sq on the

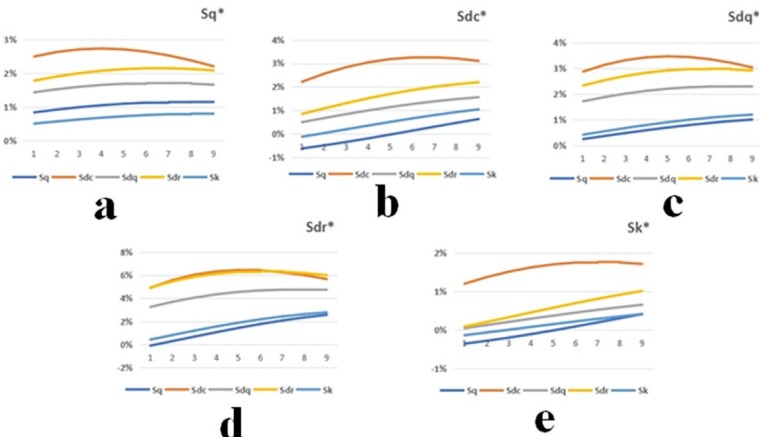

**Fig 8. Influence of unworn surface parameters on the worn surface parameters.** (a) The change rate curve of Sq*. (b) The change rate curve of Sdc*. (c) The change rate curve of Sdq*. (d) The change rate curve of Sdr*. (e) The change rate curve of Sk*.

five parameters after running-in are roughly similar. Sk has a slightly larger effect on the other parameters than Sq except the change law of Sq* (as shown in Fig 8A).

According to the scatter diagram, the change in Sdc has the greatest influence on the five parameters after running-in. Furthermore, Sdr* is most affected by the parameters before running-in. Parameter Sk has the least influence on five parameters after running-in, and Sq* is the least affected by the parameters before running-in.

## 5.2. The result of multiple parameters

The analysis of the experimental data is used by the Pearson correlation coefficient; therefore, we can study how the morphology parameters after running-in are influenced by the two most relevant surface parameters before running-in. According to Fig 10, the two strongest correlations with Sq* are Sdc and Sdq. Sdc and Sdr have the strongest correlation with Sdc*, while the two strongest correlations with parameter Sdq* are Sdc and Sdq. Two unworn parameters that have the strongest correlation with the parameter Sdr* are Sdr and Sdq. Sdc and Sdr have the strongest correlation with Sk*.

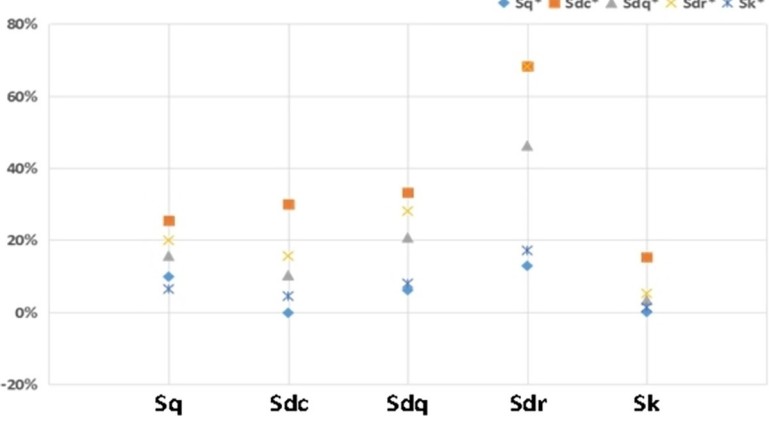

**Fig 9. The change rate of the parameters in running-in.**

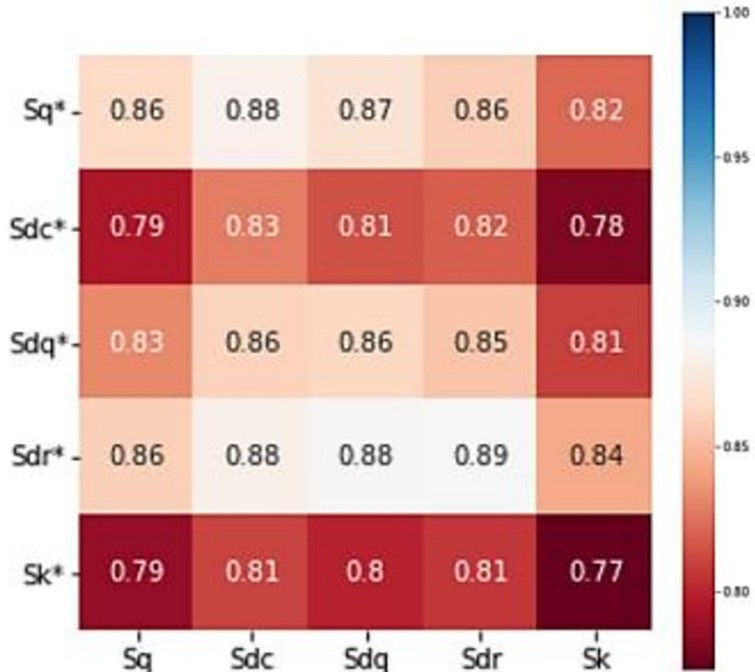

**Fig 10. Parameter correlation of the surface morphology at the beginning and end of the running-in wear process.**

Based on the correlation parameter pairs selected above, the constructed testing data are input as model predictions for analysis. The results are presented in Fig 11.

Although their relationship is nonlinear, the effect of the parameter increases before running-in on the parameters after running-in shows an obvious upward trend. The effect of Sdc on Sq* is less than that of Sdq on Sq*. Moreover, the effect of Sdc on Sdc* is less than that of Sdr on Sdc*. The effect of Sdc on Sdq* is less than that of Sdq on Sdq*, and the effect of Sdr on Sdr* is less than that of Sdq on Sdr*. The effect of Sdc on Sk* is less than that of Sdr on Sk*.

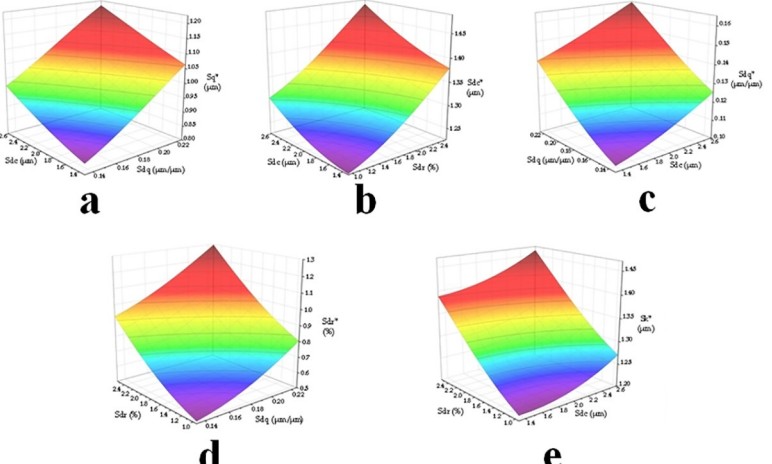

**Fig 11. Influence of parameters in running-in.** (a) The impact trend between Sq* and Sdc and Sdq. (b) The impact trend between Sdc* and Sdc and Sdr. (c) The impact trend between Sdq* and Sdc and Sdq. (d) The impact trend between Sdr* and Sdr and Sdq. (e) The impact trend between Sk* and Sdc and Sdr.

Table 4. Verification of three different working conditions.

| Serial number | Viscosity (20°C, mm²/s) | Speed (r/min) | Load (N) |
|---|---|---|---|
| 1 | 10 | 50 | 27 |
| 2 | 80 | 30 | 36 |
| 3 | 130 | 50 | 45 |

## 5.3. Result verification

The influence result above comes from the established SVM model testing. To verify its reliability, three different working conditions were set in the running-in wear experiment and are shown in Table 4.

Under the above conditions, the experimental value of worn surface morphology and the prediction value by SVM modeling are shown in Fig 12.

The result of model verification prediction presents good prediction accuracy. Considering three working conditions different from those of the running-in experiment used in model training, the prediction result in Fig 12 shows good model generalization capability.

To verify the influence result based on the new experimental data, an analysis formula was adopted and is shown as follows:

$$Z = |\frac{y_2 - y_1}{y_1}| / |\frac{x_2 - x_1}{x_1}| \tag{5}$$

$x_2$ is the unchanged parameter of the unworn surface, and $x_1$ is the changed parameter of the unworn surface. $y_2$ is the unchanged parameter of the worn surface, and $y_1$ is the changed parameter of the worn surface. In the formula, the denominator is the change rate of the parameters before running-in, and the numerator is the change rate of the parameters after running-in. The larger Z is, the larger the influence of the parameter before running-in on the parameter after running-in.

According to the data collected in the three new experiments, the influence comparison was evaluated based on the analysis formula above, and the verification of the influence result is shown in Fig 13.

Fig 13 shows that the effect of Sdc on Sq* is less than that of Sdq on Sq*. Moreover, the effect of Sdc on Sdc* is less than that of Sdr on Sdc*. The effect of Sdc on Sdq* is less than that of Sdq

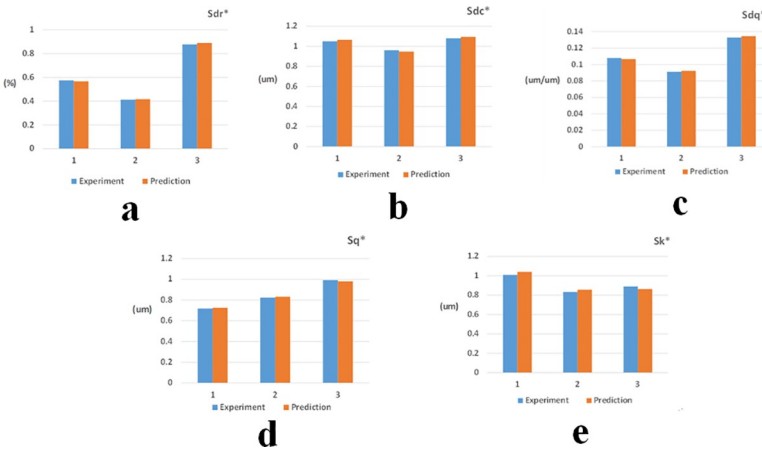

Fig 12. Verification of model prediction. (a) Sq*. (b) Sdc*. (c) Sdq*. (d) Sdr*. (e) Sk*.

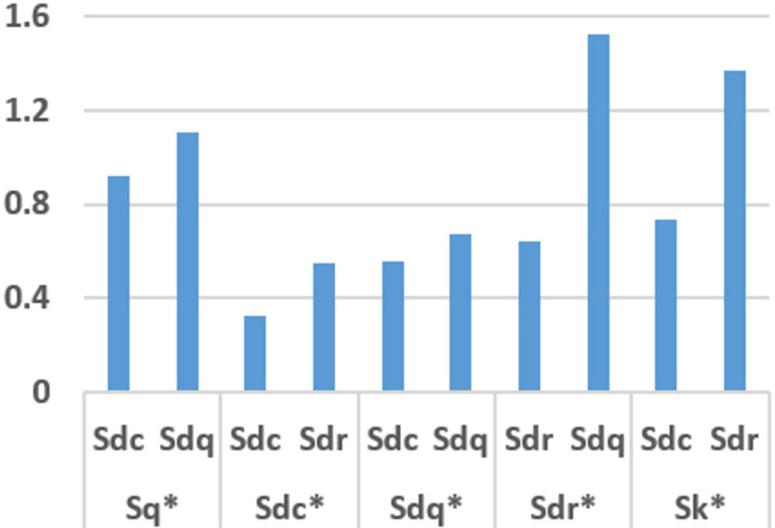

**Fig 13. Verification of the influence result.**

on Sdq*, and the effect of Sdr on Sdr* is less than that of Sdq on Sdr*. The effect of Sdc on Sk* is less than that of Sdr on Sk*. The influence verification result is consistent with the findings of multiple parameters.

In summary, the model prediction in method selection and result verification shows the same good prediction accuracy. Meanwhile, the change law of the surface parameter influence from model testing and experimental analysis is consistent. Therefore, the model and change law of the surface parameter influence can be considered effective.

## 5.4. Discussion

By studying the influence of parameters in running-in, the optimization direction of parameters on performance can be explored. There are three main functional performances considered in this paper: oil storage, support and friction performance.

When the oil storage capacity is desired, the surface can hold more lubricant in the stable wear stage. Moreover, oil storage capacity has a specific relationship with sealing performance [64]. When the surface has better support performance, the surface core morphology becomes smooth [61] and bears more load without failure. Considering the power consumption of a mechanical system, a low COF is usually desired. For this case, the lower the COF is, the better the friction performance, and the more surface morphology requirements there are.

According to the presented findings, lubricant retention involves many factors and deserves detailed discussion. The lubricant retention area should be deepened, and the main supporting height of the surface core roughness should be higher to enhance the oil storage capacity, which can improve the lubrication performance to prolong the service life of the equipment. Considering that the oil storage capacity is more affected by Svk than Sk, Svk is considered the main parameter affecting the oil storage capacity of the surface morphology. When the values of Sq and Sk on the initial surface increase, Svk* also increases, so the oil storage capacity of the surface is stronger [53]; thus, better lubrication indirectly results in a lower COF.

The oil storage capacity of the worn surface is also affected by the surface parameters Sdr and Sdq. The texture is simpler when Sdq decreases and more complex when Sdq increases. Based on the conclusions provided in sections 5.1 and 5.2, Sdr* and Sdq* can be increased by increasing Sdc, Sdq, and Sdr, which consequently improves the oil storage capacity. However,

because the Sdc and Sdr parameters of the initial surface significantly affect other parameters after running-in, increasing the Sdq of the initial surface to enhance the texture complexity after running-in is appropriate.

The surface parameter Sdc indirectly reflects the oil storage capacity of a surface, and it correlates with the sealing performance of a surface. An Sdc that is too large leads to the leakage of lubricants, and an Sdc that is too small results in poor fluid retention capacity. Considering that Sdc* has the strongest correlation with Sdc and Sdr and that Sdc* is most affected by the Sdr of the unworn surface, it is recommended that medium Sdc and Sdr parameters be chosen.

To enhance the oil storage capacity of the worn surface, it is recommended that the values of Svk*, Sk*, Sdr*, and Sdq* parameters be large, while the value of the Sdc* parameter be medium. Considering the internal impact relationship of parameters before and after running-in, it is suggested that the values of Sq, Sdq, and Sk be larger, while Sdc and Sdr be medium.

Usually, COF control involves surface morphology and lubricant retention. Considering that running-in wear is an unstable process, the rougher the surface morphology is, the greater the COF. For cooling friction pairs and carrying wear debris, the more lubricant detained, the better the lubrication. Then, a smoother surface and more lubricant will lower the COF.

When Sk* increases, the surface becomes rougher, the support performance decreases and the COF increases. When the hybrid parameter Sdq* is relatively low, the support performance of the worn surface is increased. Therefore, to improve the worn surface support performance, medium Sk* and relatively low Sdq* parameter values are necessary. Considering the change trend of the model testing result in section 5.2, smaller Sdc, Sdq and Sdr values of the unworn surface provide better support performance.

When good oil storage capacity in the stable wear stage is required, the structure of the lower part of the surface morphology should provide more space for lubricant retention. Increasing the values of Sq, Sdq and Sk will lead to this, while medium values of Sdc and Sdr are helpful.

When a lower COF in the stable wear stage is required, a lower core of the peak is required, and the main goal is to obtain a smaller Sk*. Based on the result in section 5.2, decreasing the values of Sdc and Sdr will lead to accomplishing this goal, and increasing the values of Sq and Sdq will work indirectly by providing more lubricant.

When good support performance in the stable wear stage is required, the core and slope of the peak should be lower. Based on the aforementioned research results, small values of Sk* and Sdq* of the worn surface are desirable. Small Sdc, Sdq and Sdr values of the initial surface will lead to attaining this.

Therefore, the results are useful for providing basic guidance for surface morphology optimization for performance design. Furthermore, the extension of other surface parameters requires further detailed exploration.

## 6. Conclusions

In this paper, models were established based on the correlation of surface parameters in the running-in wear stage. A comparative analysis of machine learning techniques was presented, and SVM modeling was found to be a suitable running-in wear simulation method. According to the established running-in model prediction, the mutual influence between the parameters in running-in was obtained. To obtain a desirable surface performance in the stable wear stage after running-in, the optimization of the surface morphology during running-in is explored. When better oil storage capacity is required, Sq, Sdq and Sk should be increased while setting

Sdc and Sdr as medium. When a lower COF is required, Sdc and Sdr should be decreased while Sq and Sdq should be increased. When better support performance is required, the recommendation is to increase Sdc, Sdq, and Sdr. These conclusions can provide a reference for improving the specific performance of the surface morphology. In addition, the method provides a solution to establish linkages between the design of unworn surfaces and the performance of worn surfaces after running-in. Furthermore, the conclusions fill the gap in quality monitoring of lifecycles.

## Author Contributions

**Data curation:** Guangyuan Ge.

**Formal analysis:** Fenfen Liu.

**Funding acquisition:** Gengpei Zhang.

**Investigation:** Fenfen Liu.

**Methodology:** Fenfen Liu.

**Project administration:** Gengpei Zhang.

**Resources:** Gengpei Zhang.

**Software:** Guangyuan Ge.

**Supervision:** Gengpei Zhang.

**Validation:** Guangyuan Ge.

**Visualization:** Guangyuan Ge.

**Writing – original draft:** Guangyuan Ge, Fenfen Liu.

**Writing – review & editing:** Gengpei Zhang.

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
