## [Decision Letter · Decision Letter 0]

25 Jun 2021

PONE-D-21-17072

Optimization of Running-in surface morphology parameters based on Auto-ML model

PLOS ONE

Dear Dr. Zhang,

Thank you for submitting your manuscript to PLOS ONE. After careful consideration, we feel that it has merit but does not fully meet PLOS ONE’s publication criteria as it currently stands. Therefore, we invite you to submit a revised version of the manuscript that addresses the points raised during the review process.

Based on reviewers comments authors are suggested to revise the paper and submit.

The authors need to do thorough proofreading of this paper. Some phrases don't make sense and there are lots of typos and wrong use of prepositions.

We look forward to receiving your revised manuscript.

Kind regards,

Rajesh Kaluri, Ph.D

Academic Editor

PLOS ONE

Journal Requirements:

1. Please ensure that your manuscript meets PLOS ONE's style requirements, including those for file naming. The PLOS ONE style templates can be found athttps://journals.plos.org/plosone/s/file?id=wjVg/PLOSOne_formatting_sample_main_body.pdf and https://journals.plos.org/plosone/s/file?id=ba62/PLOSOne_formatting_sample_title_authors_affiliations.pdf

Additional Editor Comments (if provided):

Based on reviewers comments authors are suggested to revise the paper and submit

Reviewers' comments:

Reviewer's Responses to Questions

**Comments to the Author**

1. Is the manuscript technically sound, and do the data support the conclusions?

Reviewer #1: Yes

Reviewer #2: Partly

2. Has the statistical analysis been performed appropriately and rigorously? 

Reviewer #1: Yes

Reviewer #2: Yes

3. Have the authors made all data underlying the findings in their manuscript fully available?

Reviewer #1: Yes

Reviewer #2: Yes

4. Is the manuscript presented in an intelligible fashion and written in standard English?

Reviewer #1: Yes

Reviewer #2: No

5. Review Comments to the Author

Reviewer #1: -Authors should improve the overall readability of the paper.

- There should a related work section showing the latest referencs and comparison of this study with existing ones.

- - There are many typos and grammatical mistakes in the entire paper.

- All the key terms of the equations must be mentioned

- The authors should further add explanation about research method.

- What are the evaluations used for the verification of results?

- Major contribution was not clearly mentioned in the conclusion part

- For ANN, Authors can cite the following paper.

1) DeepAMD: Detection and identification of Android malware using high-efficient Deep Artificial Neural Network, Future Generation Computer Systems 115, 844-856

- for bagging authors can cite below paper:

A Novel PCA-Firefly Based XGBoost Classification Model for Intrusion Detection in Networks Using GPU

Reviewer #2: While the overall organization of the manuscript seems clear and the collection of empirical data is also good, however the manuscript requires the following changes

Introduction needs to explain the main contributions of the work more clearly

Authors should improve on their language and grammar. I advise seeking help from a native English reader.

Literature review techniques have to be strengthened by including the issues in the current system and how the author proposes to overcome the same.

Hyperlinks are not working for the equations and figures

The below article can be cited if needed

Analysis of dimensionality reduction techniques on big data

6. PLOS authors have the option to publish the peer review history of their article (what does this mean?). If published, this will include your full peer review and any attached files.

Reviewer #1: No

Reviewer #2: No

---

## [Author Response · Author response to Decision Letter 0]

22 Jul 2021

The authors wish to thank the Editors and the anonymous reviewers very much for their valuable comments and suggestions, which leads to significant improvement of the quality and presentation of the manuscript. In the following, we will explain how the reviewer' comments have been considered in the revision.

Suggestions from Chief editor:

1. The authors need to do thorough proofreading of this paper. Some phrases don't make sense and there are lots of typos and wrong use of prepositions.

We are extremely sorry for the typos and grammatical errors. The manuscript has been carefully checked by authors and revised by a professional English language editing company to improve the grammar and readability.

2. The authors should emphasize the difference between other methods to clarify the position of this work further. 

We highly appreciate the editor for this comment. We have accordingly revised the introduction, discussion and conclusions in the revised manuscript. Through comparative analysis of literatures in relevant fields, we have clarified the positioning of this work. At the same time, we have added relevant content to clarify the rationality of the method selection.

Reviewers' comments:

Reviewer #1: 

1. Authors should improve the overall readability of the paper.

Author response: The manuscript has been carefully checked by authors and revised by a professional English language editing company to improve readability.

2. There should a related work section showing the latest references and comparison of this study with existing ones.

Author response: We are extremely grateful for this comment. We have rewritten the content of the introduction part. At the same time, we have added and deleted some references in the revised manuscript. Through comparative analysis of literatures in relevant fields, we have clarified the positioning of this work.

3. There are many typos and grammatical mistakes in the entire paper.

Author response: We are extremely sorry for the typos and grammatical errors. The manuscript has been carefully checked and revised.

4. All the key terms of the equations must be mentioned

Author response: Thank you for your comment. We have included the relevant content in the revised manuscript.

5. The authors should further add explanation about research method.

Author response: Thank you for your valuable advice. We have added relevant content to clarify the rationality and correctness in 4.3. Method selection.

6. What are the evaluations used for the verification of results?

Author response: We appreciate the reviewer for the comment and have supplemented the relevant content (Section 5.3. Result verification) related to the verification of results in the revised manuscript. To verify the rationality of this model, three different working conditions were set in the running-in wear experiment. The prediction of model verification presents good prediction accuracy and shows good model generalization capability, and the results from model testing and verification are consistent.

7. Major contribution was not clearly mentioned in the conclusion part

Author response: Thank you for your comment. We have revised the relevant content in the conclusions of the revised manuscript. We have clarified the main contribution of the article in the revised manuscript. This work has provided a solution to establish linkages between design of unworn surface and performance of worn surface after running-in. Furthermore, the conclusions fill the gap in quality monitoring of lifecycle.

8. For ANN, Authors can cite the following paper.

1) DeepAMD: Detection and identification of Android malware using high-efficient Deep Artificial Neural Network, Future Generation Computer Systems 115, 844-856

- for bagging authors can cite below paper:

A Novel PCA-Firefly Based XGBoost Classification Model for Intrusion Detection in Networks Using GPU

Author response: Thank you for your kind advice. We have supplemented these references in the revised manuscript.

Reviewer #2:

1. Introduction needs to explain the main contributions of the work more clearly.

Author response: Thank you for pointing this out. We have rewritten the content of the introduction section and added some content to explain the main contributions of this work more clearly.

2. Authors should improve on their language and grammar. I advise seeking help from a native English reader.

Author response: We are extremely sorry for the typos and grammatical errors. We have carefully checked this article and updated the manuscript by inviting a professional English language editing service to revise it carefully in order to avoid language errors and typos.

3. Literature review techniques have to be strengthened by including the issues in the current system and how the author proposes to overcome the same.

Author response: We appreciate the reviewer for this comment. We have updated the manuscript by rewriting the introduction. We have introduced the importance of running-in wear research and analyzed the present situation and limitations of related research. Furthermore, through the analysis and research of the relevant literatures in the relevant fields, we have proposed to apply machine learning methods to the field of running-in wear. 

4. Hyperlinks are not working for the equations and figures

Author response: Thank you for your comment. We have amended the relevant parts in the revised manuscript.

5. The below article can be cited if needed

Analysis of dimensionality reduction techniques on big data

Author response: Thank you for your kind advice. We have supplemented this reference in the revised manuscript.

---

## [Decision Letter · Decision Letter 1]

24 Aug 2021

PONE-D-21-17072R1

Optimization of running-in surface morphology parameters based on the AutoML model

PLOS ONE

Dear Dr. Zhang,

Thank you for submitting your manuscript to PLOS ONE. After careful consideration, we feel that it has merit but does not fully meet PLOS ONE’s publication criteria as it currently stands. Therefore, we invite you to submit a revised version of the manuscript that addresses the points raised during the review process.

We look forward to receiving your revised manuscript.

Kind regards,

Seyedali Mirjalili

Academic Editor

PLOS ONE

Journal Requirements:

Reviewers' comments:

Reviewer's Responses to Questions

**Comments to the Author**

1. If the authors have adequately addressed your comments raised in a previous round of review and you feel that this manuscript is now acceptable for publication, you may indicate that here to bypass the “Comments to the Author” section, enter your conflict of interest statement in the “Confidential to Editor” section, and submit your "Accept" recommendation.

Reviewer #1: All comments have been addressed

Reviewer #2: All comments have been addressed

2. Is the manuscript technically sound, and do the data support the conclusions?

Reviewer #1: Yes

Reviewer #2: Yes

3. Has the statistical analysis been performed appropriately and rigorously? 

Reviewer #1: Yes

Reviewer #2: Yes

4. Have the authors made all data underlying the findings in their manuscript fully available?

Reviewer #1: Yes

Reviewer #2: Yes

5. Is the manuscript presented in an intelligible fashion and written in standard English?

Reviewer #1: Yes

Reviewer #2: Yes

6. Review Comments to the Author

Reviewer #1: The authors have addressed almost all my suggestions. I would like to accept this paper.

Reviewer #2: i didnot find the responses for the given queries so am suggesting you to highlight the responses for each query in the manuscript where you have done the changes

7. PLOS authors have the option to publish the peer review history of their article (what does this mean?). If published, this will include your full peer review and any attached files.

Reviewer #1: No

Reviewer #2: No

---

## [Author Response · Author response to Decision Letter 1]

27 Aug 2021

The authors wish to thank the Editors and the anonymous reviewers very much for their valuable comments and suggestions, which leads to significant improvement of the quality and presentation of the manuscript. In the following, we will explain how the journal requirements and reviewer' comments have been considered in the revision.

Journal Requirements

Author response: Thank you for pointing this out, we have reviewed the reference list to ensure that it is complete and correct. To be more specific, we have modified the format of the reference according to the corresponding template, and the modified sections are highlighted in yellow. 

Reviewers' comments:

Reviewer #2: I did not find the responses for the given queries so am suggesting you to highlight the responses for each query in the manuscript where you have done the changes

Author response: According to the reviewer’s suggestion, we have highlighted the responses for each query in the manuscript annotated next to the manuscript. Besides, we integrated the comments of the reviewers and gave detailed answers to each comment, clearly marking the position of the modification. As follows:

1.Introduction needs to explain the main contributions of the work more clearly.

Author response: Thank you for pointing this out. We have rewritten the content of the introduction section and added some content to explain the main contributions of this work more clearly.

The specific location is as follows: Lines 88 to 101

Based on the analysis above, the main ML methods are applied in this paper. First, the training data of the model come from the running-in wear experiment, and the detailed description is provided in sections 3.1 and 3.2. Then, surface parameter correlation analysis is conducted for the dimension reduction of model input/output, and the selection results are shown in section 4.2. According to the theory described in sections 2.1-2.5, numerous models are established in section 4.1 and evaluated in section 4.3 for method selection. To understand the influence of surface morphology in running-in, the optimal model based on the selected method is tested, the detailed operation is presented in section 4.4, and the results are shown in sections 5.1 and 5.2. To verify the model and testing result, extra experimental data based on new working conditions are obtained. The data acquisition and verification are shown in section 5.3. Finally, through the influence between the parameters, the optimization guidance of surface morphology parameters in practice is presented in section 5.4.

Based on the sequence mentioned above, this paper attempts to establish linkages between the design of unworn surfaces and the performance of worn surfaces after running-in.

2.Literature review techniques have to be strengthened by including the issues in the current system and how the author proposes to overcome the same.

Author response: We appreciate the reviewer for this comment. We have updated the manuscript by rewriting the introduction. We have introduced the importance of running-in wear research and analyzed the present situation and limitations of related research. Furthermore, through the analysis and research of the relevant literatures in the relevant fields, we have proposed to apply machine learning methods to the field of running-in wear. 

The specific location is as follows: Lines 60 to 87

In 2015, to simulate the surface profile roughness of materials, Agrawal et al. [24] established multiple regression, random forests, and quantile regression models. In 2017, Wu, D et al. used RFs and support vector regression for tool wear prediction [25] and used RFs to predict tool wear in dry milling operations [26]. To distinguish the difference of ML techniques for predicting tool wear during milling, Wu et al. [27] compared the performance of ANN, SVM regression, and RF. In 2018, Bustillo et al. [28] used artificial regression trees, multilayer perceptrons, radial basis networks, and RFs to predict the surface roughness, quality loss, and flatness deviation of machined surfaces under different processing parameters [29]. To achieve real-time prediction of surface roughness deviation, Pimenov et al. [30] evaluated ML methods such as the RF, regression trees, the standard multilayer perceptron, and the radial basis function. Gouarir A et al. proposed a tool wear prediction system based on the convolutional neural network method [31]. In 2019, to realize real-time and accurate monitoring of tool wear during processing, Kong D et al. used traditional methods such as an ANN and an SVM to achieve tool wear prediction and then proposed tool wear prediction models based on core major component analysis and related vector machines [32]. Altay O et al. used linear regression, an SVM, and Gauss process regression to predict wear quantities [33]. In 2020, Thankachan T et al. predicted and analyzed the dry sliding wear rate on novel copper-based surface composites based on the source feed forward back propagation model of ANNs [34]. Cheng M et al. established an intelligent prediction model by using the support vector regression method [35]. In 2021, Alajmi et al. used Gaussian process regression, an SVM and an ANN to establish a cutting force prediction model of alloy steel [36].

Many methods based on ML have been widely applied in the field of tool wear prediction. Some researchers have used various ML methods to predict surface roughness, wear rate and wear volume in wear research. However, ML methods are barely used to study the changing laws of the surface morphology in the running-in wear process. Previous studies have shown that ML methods achieve great superiority in the field of wear. Thus, if ML methods can be used to conduct research in the field of running-in wear, it will greatly reduce the complexity of the research and improve the prediction accuracy.

3.All the key terms of the equations must be mentioned

Author response: Thank you for your comment. We have included the relevant content in the revised manuscript.

The specific location is as follows: Lines 248 to 252

whereis the result of surface morphology parameters predicted by ML after data preprocessing. This result can become the prediction results of surface morphology parameters by employing inverse data processing. is the working condition parameter of surface morphology after data processing, andis surface morphology parameters before running-in and after data preprocessing.

4.The authors should further add explanation about research method.

Author response: Thank you for your valuable advice. We have added relevant content to clarify the rationality and correctness in 4.3. Method selection.

The specific location is as follows: Lines 319 to 332

For investigations regarding Sdc and Sk as running-in parameters, conclusions obtained by TPOT in this paper can be used for reference. The SVM is recommended as the modeling method when comprehensive parameter modeling analysis is required.

At the same time, ML approaches are learning methods that rely on data. ML mainly analyzes sample data, discovers laws from data and builds a suitable model. This model can be used to predict unknown sample data and unobservable sample data. Theoretically, the unique advantages of the SVM show better modeling effects in the process of running-in wear modeling: 1. the process of an SVM searching for support vectors is a convex optimization problem; 2. An SVM transforms a nonlinear problem into a linear problem through a kernel function, which can weaken the complexity of the original problem; 3. The SVM extracts support vector data from the training data and performs weight processing on the support vector data. If the amount of training data is small, all training data can be used as support vectors to build the model, which greatly reduces the number of training samples. Furthermore, SVMs use the principle of structural risk minimization, which makes the model stronger in generalization.

5.What are the evaluations used for the verification of results?

Author response: We appreciate the reviewer for the comment and have supplemented the relevant content (Section 5.3. Result verification) related to the verification of results in the revised manuscript. To verify the rationality of this model, three different working conditions were set in the running-in wear experiment. The prediction of model verification presents good prediction accuracy and shows good model generalization capability, and the results from model testing and verification are consistent.

The specific location is as follows: Lines 416 to 452

5.3. Result verification

The influence result above comes from the established SVM model testing. To verify its reliability, three different working conditions were set in the running-in wear experiment and are shown in Table 4.

TABLE 4

VERIFICATION OF THREE DIFFERENT WORKING CONDITIONS

Serial number Viscosity (20◦C, mm2/s) Speed (r/min) Load (N)

1 10 50 27

2 80 30 36

3 130 50 45

Under the above conditions, the experimental value of worn surface morphology and the prediction value by SVM modeling are shown in Fig 11.

Fig 11. Verification of model prediction. (a) Sq*. (b) Sdc*. (c) Sdq*. (d) Sdr*. (e) Sk*.

The result of model verification prediction presents good prediction accuracy. Considering three working conditions different from those of the running-in experiment used in model training, the prediction result in Fig 11 shows good model generalization capability.

To verify the influence result based on the new experimental data, an analysis formula was adopted and is shown as follows:

 (5)

x2 is the unchanged parameter of the unworn surface, and x1 is the changed parameter of the unworn surface. y2 is the unchanged parameter of the worn surface, and y1 is the changed parameter of the worn surface. In the formula, the denominator is the change rate of the parameters before running-in, and the numerator is the change rate of the parameters after running-in. The larger Z is, the larger the influence of the parameter before running-in on the parameter after running-in.

According to the data collected in the three new experiments, the influence comparison was evaluated based on the analysis formula above, and the verification of the influence result is shown in Fig 12.

Fig 12. Verification of the influence result.

Fig 12 shows that the effect of Sdc on Sq* is less than that of Sdq on Sq*. Moreover, the effect of Sdc on Sdc* is less than that of Sdr on Sdc*. The effect of Sdc on Sdq* is less than that of Sdq on Sdq*, and the effect of Sdr on Sdr* is less than that of Sdq on Sdr*. The effect of Sdc on Sk* is less than that of Sdr on Sk*. The influence verification result is consistent with the findings of multiple parameters.

In summary, the model prediction in method selection and result verification shows the same good prediction accuracy. Meanwhile, the change law of the surface parameter influence from model testing and experimental analysis is consistent. Therefore, the model and change law of the surface parameter influence can be considered effective.

6.Major contribution was not clearly mentioned in the conclusion part

Author response: Thank you for your comment. We have revised the relevant content in the conclusions of the revised manuscript. We have clarified the main contribution of the article in the revised manuscript. This work has provided a solution to establish linkages between design of unworn surface and performance of worn surface after running-in. Furthermore, the conclusions fill the gap in quality monitoring of lifecycle.

The specific location is as follows: Lines 512 to 524

In this paper, models were established based on the correlation of surface parameters in the running-in wear stage. A comparative analysis of machine learning techniques was presented, and SVM modeling was found to be a suitable running-in wear simulation method. According to the established running-in model prediction, the mutual influence between the parameters in running-in was obtained. To obtain a desirable surface performance in the stable wear stage after running-in, the optimization of the surface morphology during running-in is explored. When better oil storage capacity is required, Sq, Sdq and Sk should be increased while setting Sdc and Sdr as medium. When a lower COF is required, Sdc and Sdr should be decreased while Sq and Sdq should be increased. When better support performance is required, the recommendation is to increase Sdc, Sdq, and Sdr. These conclusions can provide a reference for improving the specific performance of the surface morphology. In addition, the method provides a solution to establish linkages between the design of unworn surfaces and the performance of worn surfaces after running-in. Furthermore, the conclusions fill the gap in quality monitoring of lifecycles.

7. Authors should improve the overall readability of the paper.

Author response: The manuscript has been carefully checked by authors and revised by a professional English language editing company to improve readability, and the modified sections are highlighted in yellow.

---

## [Editor Report · Decision Letter 2]

13 Sep 2021

Optimization of running-in surface morphology parameters based on the AutoML model

PONE-D-21-17072R2

Dear Dr. Zhang,

We’re pleased to inform you that your manuscript has been judged scientifically suitable for publication and will be formally accepted for publication once it meets all outstanding technical requirements.

Kind regards,

Seyedali Mirjalili

Academic Editor

PLOS ONE
---

## [Editor Report · Acceptance letter]

23 Sep 2021

PONE-D-21-17072R2 

Optimization of running-in surface morphology parameters based on the AutoML model 

Dear Dr. Zhang:

I'm pleased to inform you that your manuscript has been deemed suitable for publication in PLOS ONE. Congratulations! Your manuscript is now with our production department. 

Kind regards, 

on behalf of

Prof. Seyedali Mirjalili 

Academic Editor

PLOS ONE